# A model of colour appearance based on efficient coding of natural images

**Jolyon Troscianko**[ID][1]*, **Daniel Osorio**[ID][2]

**1** Centre for Ecology & Conservation, University of Exeter, Penryn, United Kingdom, **2** School of Life Sciences, University of Sussex, Brighton, United Kingdom

* jt@jolyon.co.uk

**Data Availability Statement:** An implementation of the SBL model is provided here: https://www.biorxiv.org/content/biorxiv/early/2022/02/23/2022.02.22.481414/DC1/embed/media-1.zip?download=true.

## Abstract

An object's colour, brightness and pattern are all influenced by its surroundings, and a number of visual phenomena and "illusions" have been discovered that highlight these often dramatic effects. Explanations for these phenomena range from low-level neural mechanisms to high-level processes that incorporate contextual information or prior knowledge. Importantly, few of these phenomena can currently be accounted for in quantitative models of colour appearance. Here we ask to what extent colour appearance is predicted by a model based on the principle of coding efficiency. The model assumes that the image is encoded by noisy spatio-chromatic filters at one octave separations, which are either circularly symmetrical or oriented. Each spatial band's lower threshold is set by the contrast sensitivity function, and the dynamic range of the band is a fixed multiple of this threshold, above which the response saturates. Filter outputs are then reweighted to give equal power in each channel for natural images. We demonstrate that the model fits human behavioural performance in psychophysics experiments, and also primate retinal ganglion responses. Next, we systematically test the model's ability to qualitatively predict over 50 brightness and colour phenomena, with almost complete success. This implies that much of colour appearance is potentially attributable to simple mechanisms evolved for efficient coding of natural images, and is a well-founded basis for modelling the vision of humans and other animals.

## Author summary

An object's brightness and colour are not just due to its own surface properties, but also depend on the colours and patterns of its surrounds. We set out to develop a computational model that could predict colour appearance based on the principle of efficient coding. This takes into account the fact that neural bandwidth is limited (e.g. the fastest rate a neurone can fire might only be ten times its lowest rate), and that none of this valuable bandwidth should be wasted when coding information across different spatial scales in a typical natural scene. We next combined these principles with contrast sensitivity functions (because contrast detection thresholds vary with spatial scale), and used either psychophysical or neurophysiological data to estimate the bandwidth for humans/primates. When we tested the model against a bank of visual phenomena (illusions) we found that

**Funding:** JT was funded by a NERC IRF (NE/
P018084/1). The funder had no role in study
design, data collection and analysis, decision to
publish, or preparation of the manuscript.

**Competing interests:** The authors have declared
that no competing interests exist.

the model was able to predict the direction of almost all phenomena. Our model is surprisingly simple and generalisable, with no free parameters, and would be explained by low-level feed-forward neural architecture. This suggests that many complex visual phenomena–that have often attributed to high-level processes–could arise as artefacts of limited bandwidth and efficient coding, offering valuable avenues for future research.

This is a *PLOS Computational Biology* Methods paper.

## Introduction

The colour and lightness of objects cannot be recovered directly from the retinal image of a scene, but depend upon neural processing by low-level spatial filters and feature detectors along with long-range and top-down mechanisms that incorporate contextual information and prior knowledge about the visual world [1–4]. Ideally, image processing achieves lightness and colour constancy–allowing us to see colour and form veridically–but inevitably it produces visual effects and illusions, which give insight into the underlying mechanisms. Thus, the surroundings of an object affect its lightness or colour in several ways. For example, assimilation and induction effects shift appearance towards that of neighbouring colours [5], whereas simultaneous contrast increases the difference between an object and the surround, and in contrast induction the surround affects the contrast of a pattern [6,7]. The crispening effect–where contrasts close to the background level are enhanced–encompasses all three of these phenomena [8,9]. Related effects in colour vision include the Abney, Bezold–Brücke, Hunt, and Stevens effects, where colours, colourfulness and contrasts shift with saturation and brightness [10].

Neural mechanisms have been proposed to account for some of the foregoing phenomena, for example Mach Bands can be attributed to lateral inhibition [11], brightness induction to spatial filtering in the primary visual cortex [12], and colour constancy to photoreceptor adaptation [13,14] or to cortical processing [15]–but these accounts are controversial, and some effects are not easily explained [7,8,16]. Moreover, the lack of a comprehensive account of colour appearance limits the accuracy of the models that are used in design, industry and research [10,17].

Although photoreceptor adaptation and lateral inhibition do partly account for colour constancy and simultaneous contrast effects, their primary function is probably better understood as allowing the visual system to efficiently encode images of natural scenes, which have a large dynamic range and a high degree of statistical redundancy. Coding efficiency, which allows the brain to make optimal use of limited neural bandwidth and metabolic energy, is a key principle in early visual processing [18–22]; here we ask how a model based on this principle might account for colour appearance.

The optimal (maximum entropy) code for natural images, as specified by their spatial autocorrelation function (i.e. second-order image statistics), approximates a Fourier transform [23,24], which is physiologically unrealistic. Efficient codes can however be defined for circularly symmetrical Difference of Gaussian (DoG) or oriented Gabor-function filters, which respectively resemble the receptive fields of retinal ganglion cells and the simple cells of mammalian visual cortex [22,25–27]. In early studies, Laughlin and his co-workers [20,28] found that the contrast response functions and the centre-surround receptive fields of fly (*Lucilia*

*vicina*) large monopolar cell (LMC) neurons—which are directly post synaptic to the photoreceptors—produce an efficient representation of natural images for the noise present the insect's photoreceptor responses. Specifically, synaptic amplification at the receptor to LMC synapse and lateral inhibition between receptor outputs, give a neural code that quantitatively accords with the methods of histogram equalization and predictive coding that are used by data compression algorithms [28]. The centre-surround receptive fields of vertebrate retinal ganglion cells are comparable to those of fly monopolar cells [29], while the simple cells in visual cortex generate an efficient code for natural image statistics [22,30].

Our aim here is not to simulate biological vision precisely, but to model efficient coding by physiologically plausible spatial filters. We describe a Spatiochromatic Bandwidth Limited (SBL) model of early vision, which uses luminance and chromatic spatial filters at octave separations to cover the detectable range of spatial frequencies (Figs 1–3). Three parameters specify the model, namely the spatial autocorrelation function (power spectrum) of natural images, noise in the retinal signal, and the channel bandwidth–or number of distinguishable response states (Fig 1; [20]). The first of these parameters is given by image statistics, the second by physiological or psychophysical measurements, and the third is estimated from psychophysical data on the crispening effect (Fig 3A; [8]). As the model predicts colour and lightness in naturalistic images, and accounts for various visual phenomena and illusions it offers a framework for understanding neural image processing, and is a starting point for simulating colour appearance for humans and other species.

## The model

The SBL model is comparable to other models of early vision that have been proposed to account for lightness and colour perception. These include MIRAGE [31], which uses non-oriented DoG filters, the oriented difference of gaussians (ODOG) model [12], the brightness induction wavelet model (BIWaM) [32], and neurodynamical brightness induction model [33], which use orientation-sensitive filters. The SBL model differs from these predecessors in that to achieve efficient coding of natural images the gain and dynamic range (i.e. contrast response function) of neural channels vary with spatial frequency–as specified by the contrast sensitivity threshold–with gain normalised to natural scene statistics, so that on average the output has equal power in each spatial channel. These gain functions are hard-wired and feed-forward, which contrasts with divisive gain models that use feedback loops to make efficient use of neural bandwidth [34].

The model is implemented as follows (Figs 1 and 2). *i*): The image is filtered with a set of spatial filters at one octave separations. These filters are either circularly symmetrical difference of Gaussian (DoG) functions [26] or Gabor functions at four orientations [25]. The filtering process differs from convolution in that it applies a Michelson contrast to centre versus surrounds; this approach follows the method proposed by Peli [35], who demonstrated that contrast in complex scenes should be scaled relative to the local luminance. The three spectral classes of filter correspond to those in human vision, namely achromatic/luminance with centre and surround receiving the same spectral inputs, blue–yellow, and red–green with centre and surround receiving opposite spectral inputs. *ii*): The lower threshold ($\alpha$) for the filter is set by the psychophysical contrast sensitivity at the filter's centre frequency (based on contrast sensitivity functions, [CFSs], [36,37], parameters shown in Table A in S1 Supporting Information). $\alpha$ is subtracted from image contrasts, which is consistent with human psychophysics [38]. The filters' contrast response function is linear over a limited dynamic range to an upper threshold ($\beta$), which is a fixed multiple, $\varepsilon$, of $\alpha$. $\varepsilon$ corresponds to the number of contrast levels that can be encoded (i.e. channel bandwidth or response states; [20]); (Figs 1 and 2). Thus, for

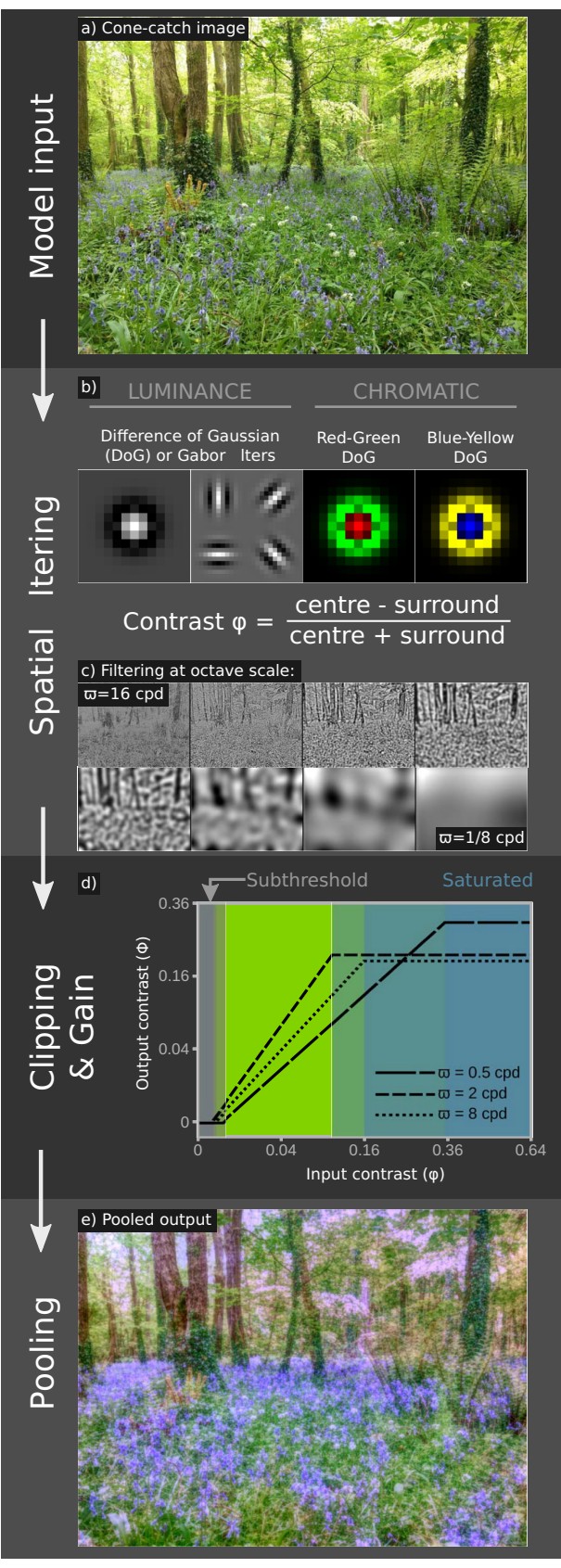

**Fig 1. Overview of the Spatiochromatic Bandwidth Limited (SBL) model.** The model uses a cone-catch image (a, S1 Appendix), which is filtered by either DoG or Gabor kernels for luminance channels, and DoG kernels for chromatic channels (b). Contrasts are converted to Michelson contrasts (c. showing luminance DoG outputs), then clipping and gain processes are applied with a bandwidth (ε) of 10 (d. Fig 2). and the spatial filters are pooled to create the output (e). Output colours are the model's internal representation and are not scaled to sRGB space. However, we note that the output image has qualities that combine the effects of an impressionist artist's take on the scene that compresses the contrasts and highlights chromatic features such as the "carpet of bluebells" that observers describe, but are much weaker in the input image. Also noteworthy is that the model would produce the same overall green scene with blue flowers irrespective of the input image's white balance.

ε = 10, the contrast saturation threshold β is 10 times the activation threshold α for each filter. As ε is equal for all channels, high sensitivity filters encode a smaller range of image contrasts than low sensitivity filters (Fig 2B and Fig B in S1 Supporting Information). We estimated ε by fitting the model to Whittle's [8] psychophysical measurement of the crispening effect (Figs 3A and 4). *iii*) Signal power in each channel is normalised to that of the filter's response to a natural scene, thereby whitening the average spatial frequency power spectrum of the output [39]. This "neural gain" at each spatial frequency remains fixed (in contrast to divisive gain models). *iv*) Filter outputs are summed to recover their representation of the original image, which can be compared to human perception of the image.

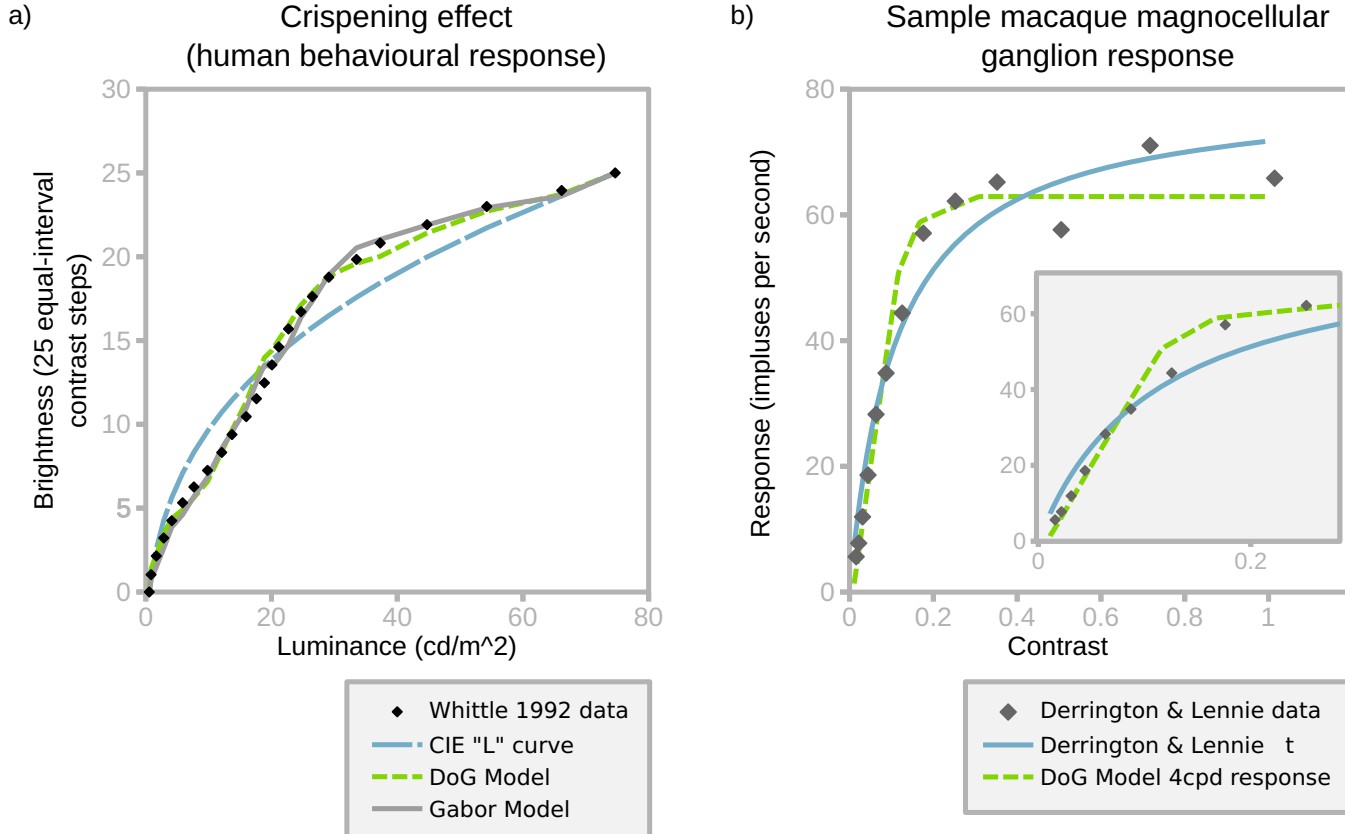

**Fig 3. Fitting the SBL model to behavioural and neurophysiological data.** a) fit to Whittle's crispening data ([8], Fig 9, "25/inc-dec/gray" treatment). Model output is scaled to the same 0–25 range. The best-fitting bandwidth (ε) for DoG filters is 15, and for Gabor (oriented) filters is 3.75, both of which result in a good fit to the raw data. The CIE L* fit specifies lightness in psychophysics and does not account for contrast [80]. b) Model fit to single ganglion response data from Derrington and Lennie ([47], Fig 11B). Fitting used a single free parameter that multiplied the arbitrary SBL model output to match neural firing responses (with zero intercept) by least-squares regression. The SBL model shows a linear contrast response and saturation point that provide a better fit than the authors' model. The inset excludes the three highest contrast values to highlight the linear relationship prior to saturation.

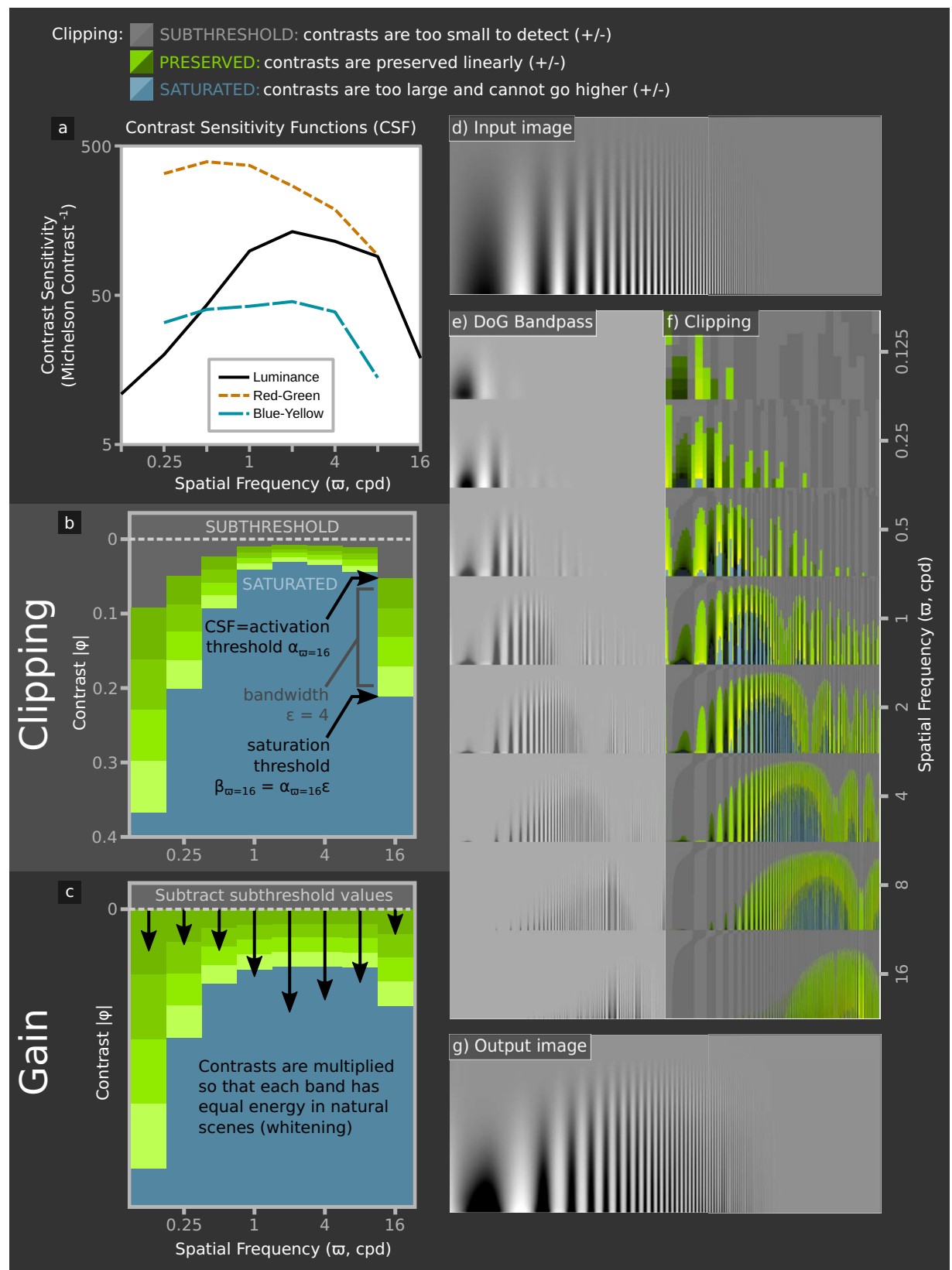

**Fig 2. Dynamic range clipping and gain adjustment by the SBL model.** a) human luminance and chromatic detection thresholds for sinewave gratings [37]. b) Clipping adjusts contrasts so that they cannot fall below the CSF at each spatial frequency ($\alpha$, SUBTHRESHOLD), or above the saturation threshold ($\beta$, SATURATED). Subthreshold contrasts are subtracted, and signals at each spatial frequency are multiplied by a gain value—denoted by arrow length in (c)—so that on average natural images have equal power at each spatial frequency (whitening). The saturation threshold is calculated from the CSF and channel bandwidth, $\varepsilon$ (4 in this example) at each spatial frequency. High and low spatial frequency channels therefore have low contrast sensitivity, but encode a large range of image contrasts, whereas intermediate spatial frequencies have high sensitivity and a low dynamic range. To demonstrate the clipping effects, we show an input image with sinewaves of different spatial frequencies and contrasts (d). (e) shows bandpass spatial filters and (f) highlights regions that are clipped or preserved. The overlap between neighbouring octaves (f) means that where contrasts are saturated for one channel, they are unlikely to be saturated for all neighbouring channels so that contrast differences are detectable even in high contrast scenes. Ultimately this shows how a system with a severely limited neural bandwidth of 15 contrast levels and peak sensitivity of ~200:1 can code for contrasts in natural scenes larger than 10,000:1. Note that the fine lines in these illustrative images suffer from moiré effects when viewed on a monitor, and we have artificially blurred the higher spatial frequencies in the input and output images to mitigate this effect. These effects were not present in the modelling, which did not use spatial frequencies that exceeded the kernel's peak sensitivity.

For the red-green and blue-yellow chromatic channels we make the assumption, consistent with neurophysiology [40,41], that the filters are less orientation selective than for luminance channels and use only DoG filters (but see [42]). The bandwidth of the red-green channel equals that of the luminance DoG signal, which produces plausible results (Fig 1 and below). However, if the blue-yellow channel has the same bandwidth ($\varepsilon$), its low contrast sensitivity (Fig 2A) means that it fails to saturate in natural scenes. We therefore reduced $\varepsilon$ to give an equal proportion of saturated pixels in natural images for red-green and blue-yellow channels.

Further details of the model are presented in S1 Supporting Information. An implementation of the SBL model is provided for use with ImageJ, a free, open-source image processing platform [43] and the micaToolbox [44,45]. The code is accessible here.

## Results

### Estimation of the bandwidth, $\varepsilon$

Channel bandwidth ($\varepsilon$) is estimated by fitting the model to human psychophysical data from Whittle's [8] investigation of the crispening effect. Whittle described how perceived lightness varies with luminance, and how contrast sensitivity depends on contrast and background luminance, by asking subjects to adjust target luminances to make equal-interval brightness series (Figs 3A and 4A). We created images simulating the viewing conditions in Whittle's experiment, including the spatial arrangement and luminance of the grey patches that he used to create perceptually uniform equal-contrast steps. Raw data (Fig 3A) were extracted from figures using WebPlotDigitiser [46]. Based on least squares fitting, $\varepsilon$ is 15 for the circularly symmetrical version of the SBF model (DoG, $R^2 = 0.994$), and 3.75 for the oriented version of the model (Gabor, $R^2 = 0.995$). These bandwidths are within the range encoded by single neurones [20,24]. Critically, the model recreates the characteristic inflection point around the background grey value. Lowering the bandwidth, and thereby increasing the proportion of saturated channels, produces a more extreme crispening effect, which suggests that crispening is due to saturation rather than to loss of contrast sensitivity with increasing contrast between targets and the background (Fig 2), which is the usual interpretation of Fechner's law [8].

Interestingly, the model with $\varepsilon$ derived from Whittle's [8] crispening data accurately predicts the responses of primate retinal ganglion cells to sinewave gratings [47] (Fig 3B). The model fit ($R^2 = 0.972$) is better than the authors' own function ($R^2 = 0.952$), although the difference in residuals is non-significant (T-test, t = 0.68, p = 0.50). Both the psychophysical crispening effect and bottom-up neural responses suggest that at around 4 cpd the saturation threshold for the human vision and macaque retinal ganglion cells ($\beta_{\varpi = 4}$) is approximately 0.2.

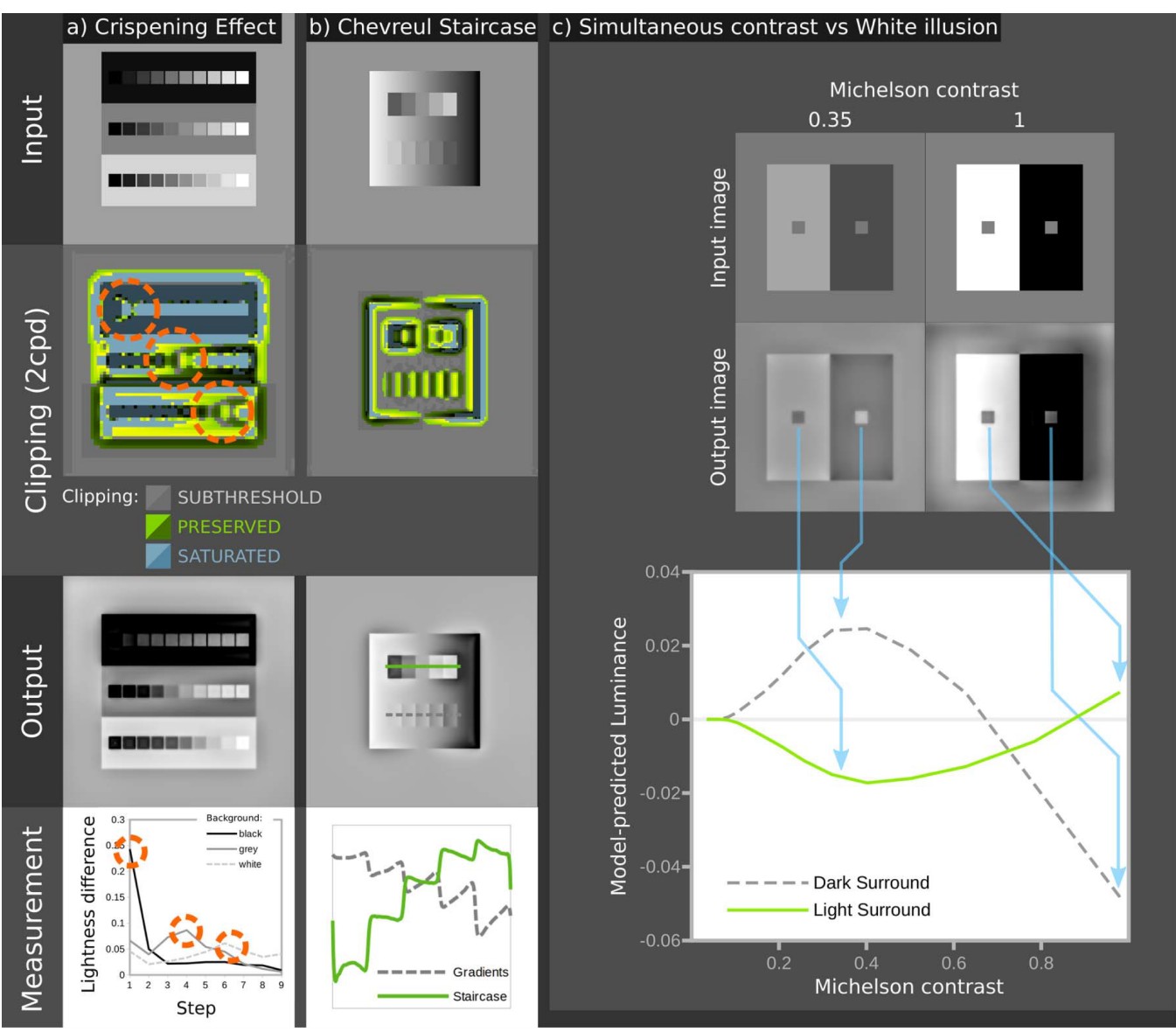

**Fig 4. Illustration of dynamic range clipping by the SBL model.** (a) for the crispening effect [8]. The three rows of grey levels are identical, with equal step sizes. Against the black background contrasts appear largest for darker squares, whereas the opposite is true for the white background. The SBL model explains this effect through saturation; contrasts near the grey level of the local surroundings are preserved (highlighted with circles), while other contrasts are saturated (blue areas adjacent to the highlighted areas). The graph at the bottom plots differences between adjacent squares in the three rows, showing higher contrasts for dark, middle and light ranges respectively. Illusions such as the Chevreul staircase (b) are also explained in part by clipping. The upper staircase appears to be a series of square steps in grey level. The lower staircase has the same grey levels, but is flipped so that its gradient matches the surround gradient. The SBL model correctly predicts that the upper staircase is seen as square steps in grey level (solid green line) while the lower staircase is a series of gradients (dashed grey line). The plot shows pixel values in arbitrary units measured along each staircase, as highlighted in the output image. The model shows that this effect arises partly because the matched gradients of the lower staircase causes local subthreshold contrasts, and because contrasts are not balanced on each side of the step. (c) Shows the effects of increasing background contrast on two identical targets. At intermediate contrasts (~0.1–0.7) the targets are predicted to show simultaneous contrast effects (the right-hand target appears lighter than the left-hand one), and at higher surround contrasts this is predicted to switch to the White illusion (spreading) effect where the right-hand target becomes darker than the left-hand one.

## Model performance

We tested the SBL model's ability to account for approximately 52 perceptual phenomena that could plausibly be explained by low-level visual mechanisms [16,48,49], first for the version

**Table 1. Summary of phenomena tested with oriented and non-oriented versions of the SBL model, with the parameters, α, β and ε fixed as explained in the text.** All phenomena were qualitatively explained to some degree. For illustrations of specific effects see the S1 Appendix.

| | | DoG model | Gabor Model |
|---|---|---|---|
| **Y:** | **Predicts effect and relevant controls** | | |
| **N:** | **Partially predicts effect, or does not predict controls** | | |
| | **Phenomenon** | **DoG model** | **Gabor Model** |
| | Crispening effect | Y | Y |
| | Contrast sensitivity | Y | Y |
| | Brightness induction/assimilation (e.g. White illusions) | Y | Y |
| | Simultaneous brightness contrast | Y | Y |
| | Illusory bars and spots (e.g. Hermann grid, Poggendorff illusion) | N | Y |
| | Contrast induction for spatial frequency, orientation, and chromatic contrast | Y | Y |
| | Colour constancy/chromatic adaptation | Y | Y |
| | Chromatic simultaneous contrast | Y | Y |
| | Chromatic assimilation | N | N |

with oriented luminance filters, and secondarily for DoG filters (chromatic filters were always non-oriented, see above). Both versions of the model correctly predict the direction of almost all effects and, where relevant, their controls (Table 1 and Fig 4 and S1 Appendix). The main exceptions were the DoG (non-oriented) model's inability to predict illusory spots and bars in the Hermann grid and Poggendorff illusions, comparatively weak performance with one control for the Chevreul staircase [50], failure in some of the brightness induction effects presented by Zaidi et al. [51] (though mostly with the comparatively weak illusions), and the enhanced assimilation of colour created by bars in patterned chromatic backgrounds [52]. Nevertheless, this performance was achieved with no free parameters (Figs 1–3), and the model can be adjusted to predict all effects.

## Discussion

The Spatiochromatic Bandwidth Limited model of colour appearance described here at least qualitatively predicts the appearance of a wide variety of images that are used to demonstrate colour and lightness perception (Table 1 and Fig 4 and S1 Appendix). These include 'illusions' that have been explained by high-level interpretations of 3D geometry, lighting, atmospherics, or mid-level principles of perceptual organisation [53,54]: for example White-Munker, shadow, Koffka ring and haze illusions. It is therefore parsimonious to suggest that many aspects of object appearance can be attributed to mechanisms adapted for–or consistent with–coding efficiency [19]. Other accounts of the same phenomena invoke specialised mechanisms (e.g. [12,55]) or top-down effects, which imply that multiple sources of sensory evidence and prior knowledge are used to infer the most likely cause of the stimulus [7,16,56,57]. Neither does the SBL model invoke light adaptation or eye movements, which implies that colour constancy is largely independent of the adaptation state of the photoreceptors–provided that they are not saturated. By comparison the models used by standard colour spaces, such as CIE LAB/CIE CAM implement the von Kries co-efficient rule [14], which assumes that photoreceptor responses are adapted to the global mean for a scene, even though chromatic adaptation is affected by both local and global colour contrasts [58]. Retinex [55] and Hunt models do normalise receptor signals to their local value [17], but the weightings of global and local factors are poorly understood. Recent work has successfully modelled some lightness phenomena by simulating edge integration with asymmetric gain for centre-on and centre-off pathways [59,60], but the mechanisms underlying colour constancy are less clear [58]. Moreover, the

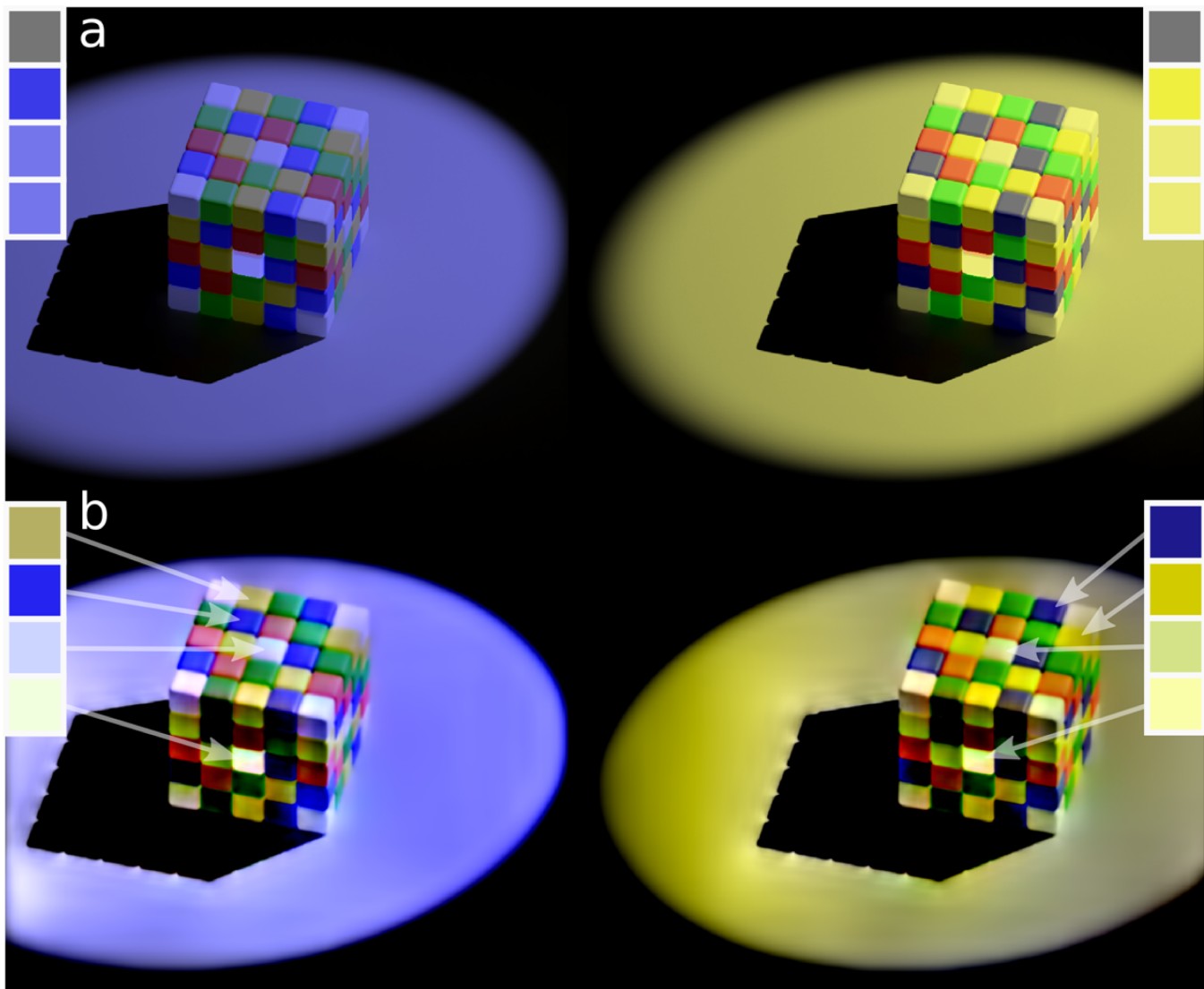

**Fig 5. The SBL model can account for colour appearance in complex naturalistic images.** (a) shows the input image (similar to the Lotto cube [81]) where the blue squares on the yellow-tinted side (right) and the yellow squares on the blue-tinted side (left) are physically the same grey (colours are shown in the squares at the top of the image). The SBL model (b) correctly predicts that the squares under both tinting regimes appear yellow and blue, rather than grey. The SBL model also predicts the powerful simultaneous contrast (or shadow) illusion present in this image whereby; the central tiles on top of the cube appear to be darker than the central tiles on the shaded side of the cube (colours shown in squares on the far left and for right hand sides).

adjustments required for colour constancy are largely complete within about 25ms [61], which is too fast for receptor adaptation, but consistent with the purely feed-forward character of the SBL model (the Rudd & Zemach model is similarly feed-forward [59,62]). Fig 5 shows how the SBL model can account for colour appearance in a naturalistic image under variable illumination. More generally, the feed-forward architecture of the SBL model explains why many other visual phenomena appear without any delay, whereas existing models require feedback loops for normalisation [12,17,32,55]. Thus, Brown and MacLeod [7] comment that the distribution of surround colours affects colour appearance almost immediately, leaving little time for feedback or adaptation. Likewise, as suggested by [6], contrast induction is explained without requiring the feedback invoked by Nassi et al. [63]. This is because, according to the SBL

model, low contrast surrounds allow all spatial bands to operate within their dynamic ranges, whereas high contrast surrounds saturate some spatial bands, resulting in under-estimates of brightness contrast or chromaticity (Figs 3A and 4). Geier and Hudák [50] argue that lateral inhibition cannot explain effects such as the Chevreul staircase, and this indeed is where the SBL model fails to replicate one of their control conditions (see S1 Appendix). However, the SBL model's performance with the other control, combined with its use of CSFs (grounding the model's performance at low spatial frequencies) does at least address many of Geier and Hudák's reasons for rejecting low-level models. Likewise, Zaidi et al. [51] show how brightness induction can be explained through t-junction based models, and while our models do not predict all effects perfectly, we show that low-level models can predict all of their effects under some conditions (e.g. White illusions and Benary cross). Future work should test the model's performance against humans quantitatively in key tasks such as this. The model also reconciles contrast constancy with a visual system that varies dramatically in contrast sensitivity and contrast gain across spatial frequencies, allowing suprathreshold contrasts to have a similar appearance at different distances [64]. Contrasts are predicted to be most constant where they are saturated across multiple spatial frequencies, e.g. where the blue regions in Fig 2F overlap. Pooling across spatial scales might explain the Abney effect, which is a shift in hue that occurs when white light is added to a monochromatic stimulus [65], because the colour stimulus may be below-threshold at some spatial bands, but above threshold for others, however we require specific data to estimate the bandwidth of chromatic channels (equivalent to Whittle's [8] luminance crispening data). As noted above (Model, Figs 1 and 2A), we assume that the bandwidth of the red-green signal equals the luminance DoG signal, but the blue-yellow signal has reduced the bandwidth, which produces plausible results when processing natural scenes, but future work should measure the chromatic bandwidth functions and determine whether the SBL model can account for the Abney effect quantitatively. Further developments of the chromatic SBL model should also investigate whether performance could be improved by modelling both single-opponent and double-opponent pathways. The latter are sensitive to both spatial frequency and orientation, and has been suggested to play a role in suprathreshold colour appearance [42]. However, we were able to simulate the same spatial-frequency/saturation effects with the non-oriented version of the SBL model (S1 Appendix).

## The Circularly Symmetric Version of the SBL Model and Animal Vision

Whereas the oriented version of the SBL model uses orientation selective achromatic filters and circularly symmetrical chromatic filters (see above), the circularly symmetrical version uses DoG filters for all channels. For the visual phenomena that we have tested the oriented version of the SBL model predicts lightness and colour at least as well as the circularly symmetrical version (Table 1). It might therefore seem logical to consider only the former, but whereas the visual systems of all animals probably have circularly symmetrical receptive fields (e.g. [28]), there is limited evidence for orientation selective cells other than in mammalian visual cortex. Also, the differences between the two versions of the SBL model seem to us to be surprisingly small. For example, both predict White effects, which might be expected to depend on orientation selective mechanisms (S1 Appendix; [12,49]), but only the oriented model correctly predicts the presence of illusory spots in the Hermann grid, and elimination of these spots in the wavy grid [66]. Similarly, the oriented version of the model predicts Koffka rings and the Chevreul staircase (Fig 4B) more accurately than the circularly symmetrical version. The bandwidth, ε, for the non-oriented filter is approximately 15, which matches neurophysiological measurements from primate retinal ganglion cells (Fig 3; [47]). By comparison the bandwidth of the oriented version is estimated to be about four-fold lower than that of the

non-oriented model, which is consistent with the low spike rates of neurons in the primary visual cortex [24]. For a given spike rate partitioning the information into multiple channels allows a correspondingly reduced integration time. The DoG model highlights an asymmetry between positive and negative contrasts in natural scenes; in order to code natural scenes efficiently and with equal bandwidth for positive (centre-on) and negative (centre-off) contrasts, our model assumes that negative contrasts require a larger dynamic range in order to use the same bandwidth efficiently (see Fig A in S1 Supporting Information). This may reflect the observed asymmetries in primate LGN ON and OFF pathways, where the OFF pathway has a larger dynamic range, and smaller receptive field size [67], and also in primate and cat cortical pathways, where the ON pathways also have a smaller dynamic range than OFF pathways [68]; our models suggest that these asymmetries may be adaptations to natural image statistics.

The SBL model is useful for non-human animals because coding efficiency is a universal principle, and contrast sensitivity functions are known for many species (Fig 2A; [69]), whereas psychophysical and neurophysiological data on visual mechanisms in non-primates is limited. Current research into non-human colour appearance typically uses the receptor noise limited (RNL) model [70,71], which also assumes that early vision is constrained by low level noise. Others have sought to control for acuity and distance dependent effects [45,69,72], but surprisingly few studies have utilised contrast sensitivity functions [73], and behavioural validation of the models is difficult [74,75]. As with human vision, the SBL model may reconcile a number of key effects. For example, in a bird (blue tit, *Cyanistes caeruleus*) chromatic discrimination thresholds depended on the contrast of the surround [74], which resembles chromatic contrast induction [7] and is simulated by the SBL model. Shadow-illusion effects have also been demonstrated in fish [76]. Aside from predicting colour appearance the SBL model highlights comparatively unexplored trade-offs in visual systems, with contrast sensitivity potentially linked to dynamic range and to other factors such as low-light vision and temporal acuity. For example, birds have poor luminance contrast sensitivity, but high temporal acuity consistent with a low neural bandwidth in the SBL model [77–79].

## Supporting information

**S1 Appendix. This file contains a comprehensive list of visual phenomena tested with both versions of the model and assessments of the models' performance.**
(PDF)

**S1 Supporting Information. This text file contains additional information regarding the model.**
(PDF)

## Acknowledgments

We thank Jenny Bosten, Nick Scott-Samuel and Roland Baddeley for their constructive feedback on earlier drafts.

## Author Contributions

**Conceptualization:** Jolyon Troscianko.

**Data curation:** Jolyon Troscianko.

**Formal analysis:** Jolyon Troscianko.

**Funding acquisition:** Jolyon Troscianko.

**Investigation:** Jolyon Troscianko, Daniel Osorio.

**Methodology:** Jolyon Troscianko, Daniel Osorio.

**Project administration:** Jolyon Troscianko, Daniel Osorio.

**Resources:** Jolyon Troscianko.

**Software:** Jolyon Troscianko.

**Supervision:** Jolyon Troscianko, Daniel Osorio.

**Validation:** Jolyon Troscianko, Daniel Osorio.

**Visualization:** Jolyon Troscianko.

**Writing – original draft:** Jolyon Troscianko, Daniel Osorio.

**Writing – review & editing:** Jolyon Troscianko, Daniel Osorio.

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
