## [Decision Letter · Decision Letter 0]

13 Aug 2022

Dear Dr Troscianko,

Thank you very much for submitting your manuscript "A model of colour appearance based on efficient coding of natural images" for consideration at PLOS Computational Biology.

As with all papers reviewed by the journal, your manuscript was reviewed by members of the editorial board and by several independent reviewers. In light of the reviews (below this email), we would like to invite the resubmission of a significantly-revised version that takes into account the reviewers' comments.

As you will see, the reviews are rather mixed. There is, however, some consensus that the work is mostly solid, but in the present form lacks the clear testable predictions that will give it sufficient novelty and biological relevance to be publishable in PLoS Computational Biology. Nonetheless, I decided to give you the opportunity to submit a substantially revised version that addresses the reviewers' concerns. In such cases it can occassionally happen that after a second round of reviews the manuscript is considered technically fine, but not of sufficient biological relevance for our journal. In such cases we frequently offer a direct transfer to PLoS One, where it then can be published often without an additional round of reviews. Of course, in this case, the decision whether you want to go this route will be with you as authors. This is not saying that this will be a likely outcome, but as I feel that this is a conceivable outcome given the current state, I wanted to communicate this possiblity clearly.

We cannot make any decision about publication until we have seen the revised manuscript and your response to the reviewers' comments. Your revised manuscript is also likely to be sent to reviewers for further evaluation.

Sincerely,

Wolfgang Einhäuser

Deputy Editor

PLOS Computational Biology

Reviewer's Responses to Questions

**Comments to the Authors:**

Reviewer #1: * General comments

- Asymmetry of ON and OFF channels

The doubling of the dynamic range of off-centre units introduces an important asymmetry (e.g., see Chichilnisky 2002). A discussion or quantification of how this asymmetry affects the results from the current model would be very interesting.

- Contrast mappings

A similar projective form for local contrast mapping was proposed by Peli (1990). Locally pooled activations also form the basis of divisive-normalisation approaches developed by Heeger and collaborators (see, e.g., 1992, 2020). Perhaps these studies could be cited and discussed.

- Feed-forward nature of the model

The model introduces considerable lateral interactions due to the divisive contrast mapping and gain equalisation. The authors could perhaps discuss how they interpret this quasi-instantaneous feedback loop. While technically implemented through a one-pass algorithm, this is not a strictly feed-forward system.

- DoG and Gabor filters

In one of the two models, the authors use both DoG and Gabor filters (for chromatic and luminance filtering). Would this mean that both these representations exist at the same hierarchical level in visual processing - which is not likely given the organisation of the mammalian early visual system (although, see also Chauhan 2020)? It would be interesting to know how the authors interpret this design decision.

- Linearity of the Michelson mapping

line 349: It is not very clear how the chromatic Michelson contrasts can be considered to distribute over a convolution. The contrast is a nonlinear operation with respect to the raw cone-inputs. Or perhaps I have misunderstood the computation that is being performed?

- Control experiments

To explore the role of various components of the model, perhaps the authors could employ control experiments. For instance, as suggested above, a very interesting manipulation would be to observe the effect of varying the degree of asymmetry between contrast increments and decrements. Another interesting manipulation is the use of other forms of contrast mapping such as Weber contrast.

- Comparisons with other models and Degree of freedom

The authors could also compare their model to other cololur appearance models such as the CIECAM02. A particularly interesting quantity would be a list of free parameters for each of the experiments that were simulated. Since the model has clearly defined phsyical interpretations for each of its parameters, this could be very informative for psychophysicists designing new studies.

* Manuscript related

- The description of the model in the main manuscript is not very clear. The authors provide a much more detailed description in the Supplementary Material which could perhaps be moved to the main text.

- Figure 2f: What do the various colours mean?

- Figure 4 : The background colour is not indicated in the Lightness Difference plots.

- Line 253: The sentence is incomplete.

- lines 369-377: The equations are not very clear. The symbol \\phi_{clipped} has not been defined before.

- line 376: It is not very clear what the authors mean by efficient coding here.

- line 347: Why is a nonlinear contrast mapping not mappable to CSFs? CSFs are nonlinear.

* References

1. Peli, E. (1990). Contrast in complex images. JOSA A, 7(10), 2032–2040.

2. Heeger, D. (1992). Normalization of cell responses in cat striate cortex. Visual Neuroscience, 9(2), 181–197.

3. Heeger, D., & Zemlianova, K. (2020). A recurrent circuit implements normalization, simulating the dynamics of V1 activity. Proceedings of the National Academy of Sciences.

4. Chichilnisky, E., & Kalmar, R. (2002). Functional Asymmetries in ON and OFF Ganglion Cells of Primate Retina. Journal of Neuroscience, 22(7).

5. Chauhan, T., Masquelier, T., & Cottereau, B. (2021). Sub-Optimality of the Early Visual System Explained Through Biologically Plausible Plasticity. Frontiers in Neuroscience, 15, 1203.

Reviewer #2: See attached review.

Reviewer #3: Attached

**Have the authors made all data and (if applicable) computational code underlying the findings in their manuscript fully available?**

Reviewer #1: Yes

Reviewer #2: Yes

Reviewer #3: **No: **

PLOS authors have the option to publish the peer review history of their article (what does this mean?). If published, this will include your full peer review and any attached files.

Reviewer #1: No

Reviewer #2: **Yes: **Michael E. Rudd

Reviewer #3: **Yes: **Qasim Zaidi
---

## [Decision Letter · Decision Letter 1]

22 Mar 2023

Dear Dr Troscianko,

Thank you very much for submitting your manuscript "A model of colour appearance based on efficient coding of natural images" for consideration at PLOS Computational Biology. As with all papers reviewed by the journal, your manuscript was reviewed by members of the editorial board and by several independent reviewers. The reviewers appreciated the attention to an important topic. Based on the reviews, we are likely to accept this manuscript for publication, providing that you modify the manuscript according to the review recommendations.

I must admit that I was a bit undecided about the paper as it stands. Without doubt the paper is now solid technically, but the links to the biological system are not too strong, they are - to use reviewer 2's words - "reasonable assumptions". In the end, I decided to go with reviewer 2's argument that it is of interest to the readership to see how far a model gets with those, so the paper is acceptable to PLoS CB, provided the final minor comments of the reviewers are addressed.

Sincerely,

Wolfgang Einhäuser

Section Editor

PLOS Computational Biology

Reviewer's Responses to Questions

**Comments to the Authors:**

Reviewer #1: The authors have responded satisfactorily to most queries.

To nitpick, I disagree that changing the order of the convolution and Michelson-contrast operation will give the same result. While it does so for the boundary conditions that the authors outline (a grey patch, or template-match inputs), the intermediate points will differ. However, the newer version of the manuscript does not raise this issue as the authors have described the exact computation they have implemented.

Reviewer #2: See attached review.

**Have the authors made all data and (if applicable) computational code underlying the findings in their manuscript fully available?**

Reviewer #1: Yes

Reviewer #2: Yes

PLOS authors have the option to publish the peer review history of their article (what does this mean?). If published, this will include your full peer review and any attached files.

Reviewer #1: No

Reviewer #2: **Yes: **Michael E. Rudd

Figure Files:

Data Requirements:

Reproducibility:

References:

---

## [Editor Report · Decision Letter 2]

20 Apr 2023

Dear Dr Troscianko,

We are pleased to inform you that your manuscript 'A model of colour appearance based on efficient coding of natural images' has been provisionally accepted for publication in PLOS Computational Biology.

Best regards,

Wolfgang Einhäuser

Section Editor

PLOS Computational Biology

Wolfgang Einhäuser

Section Editor

PLOS Computational Biology

---

## [Editor Report · Acceptance letter]

23 May 2023

PCOMPBIOL-D-22-00561R2 

A model of colour appearance based on efficient coding of natural images

Dear Dr Troscianko,

I am pleased to inform you that your manuscript has been formally accepted for publication in PLOS Computational Biology. Your manuscript is now with our production department and you will be notified of the publication date in due course.

With kind regards,

Zsofi Zombor
